# Psychological and Social Effects of Oral Health and Dental Aesthetic in Adolescence and Early Adulthood: An Observational Study

**DOI:** 10.3390/ijerph18179022

**Published:** 2021-08-27

**Authors:** Angela Militi, Federica Sicari, Marco Portelli, Emanuele Maria Merlo, Antonella Terranova, Fabio Frisone, Riccardo Nucera, Angela Alibrandi, Salvatore Settineri

**Affiliations:** 1Department of Biomedical, Dental Science and Morphological and Functional Images, University of Messina, 98100 Messina, Italy; amiliti@unime.it (A.M.); antonella.terranova@unime.it (A.T.); rnucera@unime.it (R.N.); salvatore.settineri@unime.it (S.S.); 2Department of Cognitive Sciences, Psychology, Educational and Cultural Studies (COSPECS), University of Messina, 98100 Messina, Italy; federica.sicari@unime.it (F.S.); emanuelemaria.merlo@unime.it (E.M.M.); fabio.frisone@unime.it (F.F.); 3CRISCAT (International Research Center for Theoretical and Applied Cognitive Sciences), University of Messina and Universitary Consortium of Eastern Mediterranean, Noto (CUMO) University of Messina, 98100 Messina, Italy; 4Unit of Statistical and Mathematical Sciences, Department of Economics, University of Messina, 98100 Messina, Italy; angela.alibrandi@unime.it

**Keywords:** dental aesthetics, oral health, well-being, self-confidence

## Abstract

Background: Is well known that oral health and dental aesthetic have significant effects on the sociality of human beings. The aim of the present study was to assess some aspects of oral health with possible repercussions in adolescent and youth, with particular reference to gender differences. Methods: A total of 190 subjects with female prevalence (F = 62.6%, M = 37%) and ages between 14 and 29 years old (Mean = 23.8; SD = 3.27) participated. Evaluation was carried using standardized instruments to assess quality of oral life (PIDAQ), negative impact of oral conditions (OHIP-14), and self-esteem (Rosenberg Self Esteem Scale). Correlational and difference analyses and linear regressions were performed. Results: Significant gender differences were found in terms of gender, in reference to variables such as self-confidence and convictions. Positive correlations emerged between psychological impact and social impact, aesthetic concern and social impact, convictions and self-confidence, oral health with psycho-social impact, and aesthetic concern, self-esteem with oral health. Inverse correlations emerged between psycho-social impact and self-confidence, aesthetic concern and self-confidence, oral health, and self-confidence. Multivariate linear regression indicated relations between age and psychological impact, sex and self-confidence, crooked teeth and conviction. Conclusions: The impact of oral health on the psychological well-being of young people is relevant. These factors, if considered within clinical practice, can improve the quality of life of the subject.

## 1. Introduction

Functions of body aesthetics are integral parts of personality and complexity that must be taken into account for the purposes of evaluation and the risk of failure, especially in those rapid and difficult stages of life such as adolescence and early adulthood. However, it is not always possible to highlight the relationships with psychopathologies intuitively connected, also because the adolescent is not aware of the relationships between discomfort and aesthetic alteration [1,2]. The relational value of the face is intuited by the subjects after a path of significant internalization (psychotherapy, introspection, other reflexive practices). Moreover, if the face is liked by others and by themselves with relative source of humoral stability; in contrast, as Synnott A. [3] points out, ugliness and physical deformations are also stigmatized by less recent literature [4]. The same irritability resulting from dissatisfaction may be due to manifestations of anxiety that affect treatment and the relationship with the dentist [5]. The face, through the smile, manifests emotional tones that affect one’s own and others’ esteem. Not by chance, as anthropology explains, has man resorted to architects such as masks to demonstrate an agreement between authenticity and emotions that is not always possible. Dental aesthetics can therefore significantly affect the well-being of adolescents and young adults, influencing important factors such as body representation and self-esteem. The impact of oral features on the bio-psychosocial functioning of these subjects was analyzed with particular reference to gender and age. The data emerged appears discordant; unlike Marques et al. [6], who did not show significant results, some authors [7,8] have suggested that younger people are more affected by the effect of oral blemishes on their self-esteem in the same way as women, who appear more severe in reference to their appearance [9] Dento-facial features have a strong impact, especially on the female gender, with significant consequences on the representation of the self [10], on oral health [11,12], and indirectly on the social sphere [7]. In fact, from an aesthetic–relational point of view, subjects with greater security are perceived as more attractive, and have a better social impact unlike those with oral disorders, which present a negative body image and a higher level of isolation [13]. From the above-mentioned research, it is clear that problems related to malocclusions, cavities, untreated decays, or injuries have significant consequences on oral aesthetics, affecting self-esteem, and indirectly individual well-being [14]. Above all, the presence of anomalies of the anterior incisors, more visible than the posterior teeth, can predict a negative self-perception [15,16]. These results support the hypothesis of a direct relationship between oral health and self-esteem, despite some long-term follow-ups [17,18,19] not having detected a significant relationship between the aforementioned elements. On the basis of the mentioned background, we hypothesize: significant gender differences related to the psychological impact of oral health; significant correlations between age, self-confidence, social impact, psychological impact, aesthetic concern, convictions, and oral health; relationships among predictor covariates such as age, gender, crooked and spaced teeth and outcomes such as self-confidence, psychological impact, conviction, and self-esteem, as indicated through regression analysis.

## 2. Materials and Methods

For the study, we selected 190 patients belonging to a private practice, thanks to the cooperation of students coming from the Dental School of the University of Messina. The students involved the selected subjects on the basis of their oral interventions and fully explained the nature of the study. The study protocol was self-administration. The students suggested the anonymous processing of data to the participants, parents, and tutors. All recruited subjects gave their consent before completing the protocol. With reference to minor subjects, the informed consent was signed by their parents or designed tutors. The research was conducted in accordance with the 1964 Declaration of Helsinki. The observation group consisted of 71 males and 119 females, aged between 14 and 29 years (mean: 23.8; SD: 3.27).

### 2.1. Observation Tools

The following tools were used for the evaluation of oral health and self-esteem: -Psychosocial Impact of Dental Aesthetics Questionnaire (PIDAQ) [20] for the analysis of the quality of oral life through the following subscales, which in this article was presented in the Italian version [21]: reasons to search orthodontic treatment; self-confidence from the dental point of view; social and psychological impact; aesthetic concern; and patient’s beliefs. A five-point Likert scale is used, ranging from 0 (no impact of dental aesthetics to QoL) to 5 (maximal impact of dental aesthetics) for each item.-Oral Health Impact Profile (OHIP-14) [22], which is based on the patient’s experience, focusing on the negative impact of oral conditions. Seven different dimensions are considered, here presented in the Italian version [23]: functional limitation; physical pain; psychological distress; physical deficiency; psychological disability; social incapacity; handicap.-Rosenberg Self Esteem Scale [24], which allows a global assessment of self-esteem, described in this article in the Italian version [25], through a measurement of both negative and positive feelings associated with the self.

### 2.2. Procedure

The numerical data were expressed as mean and standard deviation and the categorical variables as number and percentage. The non-parametric approach was used because examined variables were not normally distributed, as verified by the Kolmogorov–Smirnov test. The Spearman test was applied in order to evaluate the correlation between age, self-confidence, social impact, psychological impact, aesthetic concern, convictions, oral health, and self-esteem. The Mann–Whitney test was applied in order to compare all numerical variables between male and female subjects. Linear regression model was estimated in order to assess the dependence of some outcome variables (self-confidence, social impact, psychological impact, aesthetic concern, convictions, self-esteem, oral health) from age, gender, crooked teeth, spaced teeth, skeletal protrusion, as indicated by the dentist. Statistical analyses were performed using SPSS 22.0 (JMP, Milano, Italy) for Windows package. A *p*-value smaller than 0.050 was considered to be statistically significant.

## 3. Results

The results obtained in the present study are reported below. Descriptive analysis:

In order to provide a description of the sample, Table 1 reports the distributions according to gender. In Table 2 descriptive statics is reported.

To highlight directions between age, self-confidence, social impact, psychological impact, aesthetic concern, convictions, oral health, and self-esteem, Spearman correlations are reported in Table 3. Linear regression analysis instead is reported in Table 4. Significant inverse correlations were found between social impact and self-confidence, suggesting that decreasing level of self-confidence corresponds to increasing social impact level. Significant inverse correlations were also shown between psychological impact and self-confidence; thus, if psychological impact increases, self-confidence decreases and vice versa. The correlation between psychological impact and social impact was positive and significant, demonstrating that the improvement in psychological impact increases with social impact. Instead, the correlation between aesthetic concern and self-confidence was inverse. The correlation between aesthetic concern with social impact was positive and significant, demonstrating that the improvement in aesthetic concern increases with social impact. The correlation between aesthetic concern and self-confidence was instead inverse. Significant and positive correlations were found between convictions and self-confidence; thus, the improvement in convictions increases with self-confidence. Significant inverse correlations were evident between oral health and self-confidence, suggesting that the decreasing level of oral health corresponds to increasing level of self-confidence. Significant positive correlations were found between oral health and social impact, psychological impact and aesthetic concern; thus, when oral health increases, each of the three factors also increase. Finally, the correlation between self-esteem and oral health was positive and significant, demonstrating that improvement in self-esteem increases with oral health. To highlight significant differences between groups (gender and age), the Mann–Whitney test was used. Significant gender differences were found in terms of gender, in reference to variables such as self-confidence and convictions; the *p*-values of the Mann–Whitney test were respectively 0.003 for self-confidence and 0.041 for convictions. No significant differences were found in relation to age.

### Regression Analysis

Independent variables were considered in order to highlight dependences. Specifically, age, sex, and crooked teeth were considered as evident aesthetic components. Their role on psychological figures was considered as relevant for self-confidence, psychological impact, and conviction. The reported data showed significant dependences between age and psychological impact, sex (female) and self-confidence, crooked teeth and conviction. All of the above-mentioned independent variables were in significant and positive relation with the reported psychological phenomena. In line with reported differences, the role of sex was relevant in reference to self-confidence.

## 4. Discussion

The results of this study highlight how dental aesthetics and oral health can affect the psychological well-being of adolescents and young adults. This work is therefore in continuity with what the literature has shown in terms of dental aesthetics, dealing with relevant topics, and currently of a certain interest; the results found refer to empirical research and evidence. Study of the data underlines that, in particular at this stage of life, the characteristics and appearance of the face play a crucial role for adolescents and young adults, with significant consequences on self-perception [26], self-esteem [27], and quality of life [12]. Physical attractiveness can therefore have a significant impact on these factors, also affecting psycho-social functioning and individual relational abilities. In line with numerous studies [28,29,30], the data obtained from our research shows that dental aesthetics has a greater influence on the psychological well-being of the female gender, with significant repercussions on the body image and adaptation. Despite Flores et al. [31] obtaining the opposite results, within our study, significant differences have emerged with reference to gender; the dento-facial aspect affects the variables “self-confidence” and “convictions”, especially in the female population, with significant consequences on individual well-being. The first factor refers to body representation and self-perception, the second one to beliefs about oral aesthetics and how it can promote health, work, and social success. The interpretation of the data we obtained therefore emphasizes that women attribute great value to dental characteristics, which are fundamental in determining personal success; it also seems that aesthetics influence the image of the above-mentioned subjects in a significant way, conditioning the perceived aspect and quality of life. These results have also highlighted the importance of some psychological components examined within our research, contributing to the emergence of various questions regarding the relationship between analyzed factors and their possible trend. The analysis carried out highlight, in particular, the existence of an inverse relationship between self-confidence and social impact, suggesting the possibility that self-confidence has a link with the social impact related to oral aesthetics. Specifically, increased dental safety may be associated with a decrease in levels of social inhibition and this could promote improved interpersonal skills and good adaptation. These results are in line with the research of Taghavi et al. [32], who found that subjects with poor dental safety implemented coping strategies such as refraining from laughing to not show their teeth or isolating themselves from the group, with significant repercussions in the relational sphere. Through these results, it is also possible to explain the inverse correlation between aesthetic concern and social impact. As explained by Claudino D. and Traebert J. [10], positive perception of body image is connected to a lowering of the social inhibitions above-mentioned. In this sense, it is evident that self-confidence is also significantly associated with the psychological impact and the variable related to aesthetic concern; in fact, a high level of self-confidence can be linked to lower levels of stress and to a better self-perception, a guarantor of individual well-being. These data are consistent with the study by Foster Page et al. [15], whose research has shown the existence of a significant connection between dental morphology and psychological characteristics. These data also explain the positive correlation that emerged between the social and psychological impact with oral health. As highlighted in the studies by Kavaliauskienė et al. [12], high levels of satisfaction related to oral health are associated with psychosocial well-being. On the other hand, the authors themselves point out that dental well-being can prevent the appearance of anxiety-depressive symptoms, emphasizing the link between the aforementioned elements. Finally, our studies have shown that even patients’ beliefs play an important role; these were positively correlated with self-confidence. In fact, it seems that those who believe that good dental aesthetics can promote individual well-being and social success are more satisfied with their appearance. With regard to the analysis of the regressions, the study of the dependencies between the examined variables has shown that age can affect the psychological well-being of the adolescent, although in some studies, the following data are not significant [6]. In fact, it seems that younger subjects are more conditioned by the effect of oral problems on their appearance, with a significant impact on psycho-social functioning. This result can be analyzed in the studies of Kaur et al. [14], according to which during adolescence, the characteristics and appearance of the face play an important role in reference to the self-perceived aspect. Especially for younger people, social relations and well-being itself depend directly on physical attractiveness [26], so aesthetic alterations can have a direct impact on psychological components such as body representation and self-esteem, ultimately influencing the subject’s quality of life. Even gender can influence self-confidence, in the terms discussed above, while the presence of crooked teeth can have significant repercussions on the patients’ beliefs, influencing the value attributed to dental aesthetics. According to these results, in fact, misalignment of teeth can affect the individual’s appearance and socio-work success, with consequences on quality of life. Contrary to the data found in the literature [33,34,35] few trends have emerged regarding the impact of self-esteem on dental aesthetics. This limit could be due to the instrument used, so we propose applying a different one, or it is due to the relationship between the observer and the observed, not particularly profound to allow the user the real expression of their problems. The overall assessment of facial aesthetics is very complex and must take into account numerous and different factors that can affect facial and dental aesthetics that can be related both to hard and soft tissues [36,37,38]; moreover, the use of 3D low dose CT investigation allows for an extremely more accurate evaluation of the anatomical structures of the facial mass and therefore to carry out a therapeutic program that takes into account the aesthetic effects of a corrective treatment of the dento-skeletal structures [39,40]. The results of our research once again highlight the importance of dental aesthetics in the lives of adolescents and young adults, with particular reference to the female gender, emphasizing the relevance of psychological parameters in orthodontic research and clinical practice, in line with well-known literature items [41,42,43].

## 5. Conclusions

On the basis of the results of our study, it is possible to assess that:The impact of oral health and dental aesthetics on psychological well-being is relevant, especially for the female sex.In adolescents and young adults, the characteristics and appearance of the face play a crucial role, with significant consequences on self-perception, self-esteem, and quality of life.Dental aesthetics has a significant social impact; younger subjects are more conditioned by the effect of oral problems on their appearance, with a significant impact on psycho-social functioning.These factors, if considered within clinical practice, can improve the quality of life of the subject.Despite the study providing significant associations and dependencies among the considered variables, some limitations emerged. First, there was a significant difference in the presence of male and female subjects, with a consistent prevalence of female subjects. Further studies should consider pair gender groups in order to consent the extension of results to wider populations. In these terms, the number of involved subjects must be implemented. Further studies should overcome these limitations.

## Figures and Tables

**Table 1 ijerph-18-09022-t001:** Absolute frequency for percentage and categories.

Sex	Frequency	Percentage
Male	71	37
Female	119	62.6
Total	190	100

**Table 2 ijerph-18-09022-t002:** Descriptive analysis for numerical data (PIDAQ, OHIP-14, Rosenberg Self Esteem Scale).

	Mean	Standard Deviation
Age	23.8	3.27
Crooked Teeth	3.79	1.20
Spaced Teeth	3.35	1.25
Skeletal Protrusion	3.85	1.22
Indicated by the Dentist	3.58	1.34
Self-Confidence	20.28	6.06
Social Impact	12.88	5.02
Psychological Impact	11.38	4.68
Aesthetic concern	4.77	2.88
Convictions	13.84	4.34
Oral Health	11.15	9.65
Self-Esteem	17.62	2.60

**Table 3 ijerph-18-09022-t003:** Nonparametric correlations [* *p* < 0.05 level (2-tailed), ** *p* < 0.01 level (2-tailed)].

	Age	Self Confidence	Social Impact	Psychological Impact	Aesthetic Concern	Convictions	Oral Health	Self-Esteem
Age	1.00	-	-	-	-	-	-	-
Self Confidence	0.067	1.00	-	-	-	-	-	-
Social Impact	−0.097	−0.502 **	1.00	-	-	-	-	-
Psychological Impact	−0.135	−0.560 **	0.591 **	1.00	-	-	-	-
Aesthetic Concern	−0.48	−0.610 **	0.528 **	0.620 **	1.00	-	-	-
Convictions	0.116	260 **	−0.019	−0.012	−0.096	1.00	-	-
Oral Health	−0.034	−0.200 **	0.356 **	0.272 **	0.262 **	0.064	1.00	-
Self-Esteem	0.027	−0.082	0.115	−0.026	0.091	0.023	0.169 *	1.00

**Table 4 ijerph-18-09022-t004:** Linear regression analysis for numerical data, * *p* < 0.05.

	Age			Sex			Crooked Teeth		
	β	x^2^/t	*p*	β	x^2^/t	*p*	β	x^2^/t	*p*
Self Confidence	0.091	10.25	0.213	0.226	30.08	0.002 *	−0.079	−0.711	0.478
Psychological Impact	−0.154	−20.08	0.039 *	−0.021	−0.279	0.781	0.178	10.58	0.115
Conviction	0.087	10.20	0.231	0.116	10.59	0.112	0.217	10.97	0.049 *

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
