# Peer review of "Psychological and Social Effects of Oral Health and Dental Aesthetic in Adolescence and Early Adulthood: An Observational Study"

_ijerph, 2021, doi:10.3390/ijerph18179022_

Round 1
Reviewer 1 Report
Although the authors have made the suggested modifications, they have incurred in self-plagiarism, introducing bibliographic references of their own that do not contribute anything to this paper except to increase the number of citations. In view of the above, I must indicate that this paper, in my opinion, should not be published.
Here are some examples:
Militi, D.; Militi, A.; Cutrupi, M.C.; Portelli, M.; Rigoli, L.; Matarese, G.; Salpietro, D.C. Genetic basis of non syndromic hypo-dontia: A DNA investigation performed on three couples of monozygotic twins about PAX9 mutation. Eur J Paed Dent 2011 353(12); Issue 1:21-24
Portelli, M.; Militi, A.; Nucera, R.; Cicciù, M.; Gherlone, E.; Lucchese, A. Orthodontic management of missing lateral incisor by miniscrew-anchored device. Minerva Stomatol. 2016, 65, 403–411
Vitale, C.; Militi, A.; Portelli, M.; Cordasco, G.; Matarese, G. Maxillary Canine-First Premolar transposition in the permanent dentition. J. Clin. Orthod. 2009, 43, 517–524 358
Portelli, M.; Nucera, R.; Militi, A.; Matarese, G. Trattamento chirurgico-ortodontico di un primo molare mandibolare affetto dacisti follicolare Mondo Ortodontico 2009 Vol. 4 ; ISSN 0391-2000
Militi, A.; Vitale, C.; Portelli, M.; Matarese, G.; Cordasco, G. Open bite anteriore con agenesia dei secondi premolari inferiori: terapia estrattiva con utilizzo di attacchi auto leganti Mondo Ortodontico 2012;37(1):1-15 362
Portelli, M.; Matarese, G.; Militi, A.; Cordasco, G.; Lucchese, A. A proportional correlation index for space analysis in mixed dentition derived from an italian population sample European Journal of Pediatric Dentistry 2012, Vol. 13/2 113-117 364
Portelli, R. Nucera, R. Fastuca, M. Cicciù, A. Lo Giudice, A. Militi, “Use of 3D Imaging for Treatment Planning in Cases of Impacted Canines” Open Dentistry Journal 2019 Vol.13;1:137-142 366
Crupi, P.; Portelli, M.; Matarese, G.; Nucera, R.; Militi, A.; Mazza, M.; Cordasco G. Correlation between cephalic posture and facial type in patients suffering from breathing obstructive sindrome European Journal of Pediatric Dentistry 2007, Vol. 8 June 368 77—82 369
Lo Giudice, A.; Fastuca, R.; Portelli, M.; Militi, A.; Bellocchio, M.; Spinuzza, P.; Briguglio, F.; Caprioglio, A.; Nucera, R. Effects of rapid vs. slow maxillary expansion on nasal cavity dimensions in growing subjects: A methodological and reproducibility study. Eur. J. Paediatr Dent. 2017, 18, 299–304
Merlo, E.M.; Frisone, F.; Settineri,S.; Mento, C. Depression signs, Teasing and Low Self-esteem in Female Obese Adolescents: a clinical evaluation. Mediterranean Journal of Clinical Psychology 2018 Apr 27;6(1).
Mento, C.; Gitto, L.; Liotta, M.; Muscatello, M.R.; Bruno, A.; Settineri, S. Dental anxiety in relation to aggressive characteristics of patients. International Journal of Psychological Research. 2014 Jul;7(2):29-37.
Author Response
Some citations among those indicated have been removed as required. The paragraph relating to the remaining bibliographic citations has been reformulated in such a way to better explain the concept that the authors intended to express and consequently the relevance of the bibliographic citations reported in this regard.
Reviewer 2 Report
I welcome the suggested corrections
Author Response
The authors thank the reviewer for the appreciation of the corrections made to the text that have been considered satisfactory
Reviewer 3 Report
Dear Authors
Congratulations for Your hard work
Sincerely Yours
Reviewer
Author Response

(The authors gave the same response as above.)

Round 2
Reviewer 1 Report
Although the authors have eliminated many of the quotations that could be considered as self-plagiarism, they have kept some of them in a forced way. There is no justification for self-plagiarism when the statements supported by their own citations could be cited with other articles.
Author Response
Dear Reviewer,
Correct self-citation conveys the level of originality in a publication accurately and enables readers to understand the development of ideas expressed by authors. Sometimes excessive self-citation can be considered rather crass and unprofessional, and in some cases unethical. Anyhow, in some cases, also consistent self-citation may be valid and forgivable. For example, if you are in a restricted field of research, as in the present case, self-citation may be unavoidable. However, in order to comply with the corrections requested, we have further reduced the self citations present, leaving only the absolutely essential ones, reducing the total number below a minimum threshold that cannot in any way be considered objectionable. I thank you so much for the suggestions and corrections requested and for taking the time to correct our paper.
This manuscript is a resubmission of an earlier submission. The following is a list of the peer review reports and author responses from that submission.
Round 1
Reviewer 1 Report
- The present study is not novel, see examples such as 10.2147/PPA.S58971, 10.1016 / j.ajodo.2015.01.027, 10.1016 / j.ajodo.2016.03.025.
- Striking is the number of self-citations: Angela Militi 16 times, Portelli 18 times, Frisone 1 time, Merlo 1 time, Nucera 12 times, Settineri 4 times... In fact it can be stated that references 44-61 should be deleted because their content has nothing to do with the subject and it gives the feeling that they have been added to increase self-citations.
- A discrepancy is observed between the number of men and women. 71 men and 119 women. Smaller samples than other similar articles.
- No discussion of informed consents (especially given that many of them are minors).
- No distributions by age range, we cannot know if there are more adolescents or young adults in the sample....
- I should have indicated the socioeconomic level of the patients recruited since it may be a variable to be taken into account.
- A dentist could have examined the patients, as shown in similar studies: 10.1007 / s11136-013-0547-x
Reviewer 2 Report
In the Materials and Methods section, there is no description of the choice of the sample, how the participants were selected, if there was a criterion in the choice of studies and how many students participated and if they were calibrated.
Add the type of study.
The average age should be entered as a result
How were the questionnaires administered, in interview mode by the students or self-administered?
There is no approval from the ethics or departmental committee of the proposing University.
It is not specified if the data were collected anonymously and if the informed consent of the parents was requested by minors.
In the Results section the female gender is not immediately written as a significant result but only gender is spoken of
use the term gender or sex
The limitations of the study should be mentioned in the conclusions.
Reviewer 3 Report
Dear Authors
In my opinion, the work is well designed and described. It describes facts that have already been researched and confirmed. In order to improve the practical relevance of this article, I would suggest referring to studies describing the impact of aesthetic assessment on patient compliance during treatment, for example:
- Albino JE, Lawrence SD, Lopes CE, Nash LB, Tedesco LA. Cooperation of adolescents in orthodontic treatment. J Behav Med. 1991; 14: 53–70
- Woolass KF, Shaw WC, Viader PH, Lewis AS. The prediction of patient co-operation in orthodontic treatment. Eur J Orthod. 1988; 10: 235–243
- Sarul M, Antoszewska-Smith J, Park H-S. Self-perception
of smile attractiveness as a reliable predictor of increased
patient compliance with an orthodontist. Adv Clin Exp Med.
2019; 28 (12): 1633-1638.
This does not require a change in the research methodology, and will allow for taking into account the practical implications of the obtained results .
Sincerely Yours
Reviewer